# Early life stress shifts critical periods and causes precocious visual cortex development

**Janet Poplawski[1,2], Tony Montina** (ID)**[3], Gerlinde A. S. Metz** (ID)**[1]** *

**1** Canadian Centre for Behavioural Neuroscience, Department of Neuroscience, University of Lethbridge, Lethbridge, AB, Canada, **2** Institute for Genetics and Cancer, University of Edinburgh, Edinburgh, Midlothian, United Kingdom, **3** Department of Chemistry and Biochemistry, University of Lethbridge, Lethbridge, AB, Canada

* gerlinde.metz@uleth.ca

**Data Availability Statement:** Open repository data associated with this article can be found at: https://figshare.com/projects/Early_Life_Stress_Shifts_Critical_Periods_and_Causes_Precocious_Visual_Cortex_Development/228891.

## Abstract

The developing nervous system displays remarkable plasticity in response to sensory stimulation during critical periods of development. Critical periods may also increase the brain's vulnerability to adverse experiences. Here we show that early-life stress (ELS) in mice shifts the timing of critical periods in the visual cortex. ELS induced by animal transportation on postnatal day 12 accelerated the opening and closing of the visual cortex critical period along with earlier maturation of visual acuity. Staining of a molecular correlate that marks the end of critical period plasticity revealed premature emergence of inhibitory perineuronal nets (PNNs) following ELS. ELS also drove lasting changes in visual cortex mRNA expression affecting genes linked to psychiatric disease risk, with hemispheric asymmetries favoring the right side. NMR spectroscopy and a metabolomics approach revealed that ELS was accompanied by activated energy metabolism and protein biosynthesis. Thus, ELS may accelerate visual system development, resulting in premature opening and closing of critical period plasticity. Overall, the data suggest that ELS desynchronizes the orchestrated temporal sequence of regional brain development potentially leading to long-term functional deficiencies. These observations provide new insights into a neurodevelopmental expense to adaptive brain plasticity. These findings also suggest that shipment of laboratory animals during vulnerable developmental ages may result in long lasting phenotypes, introducing critical confounds to the experimental design.

## Introduction

Neuronal circuits of the developing brain demonstrate remarkable plasticity in response to early-life experiences. Regionally defined discrete periods of heightened sensitivity enable structural modification that become essentially irreversible beyond a certain age [1–4]. These transient critical periods of plasticity allow for an experience-dependent, adaptive optimization of neuronal connectivity [5]. Through critical periods, early auditory [6, 7] and visual [8] stimulation can define preference behaviours in later life that are vital to social and cultural functioning.

**Funding:** This research was funded by the Natural Sciences and Engineering Research Council of Canada (NSERC) Grants #05628 and #00031 (GM). JP was supported by an Alberta Innovates Graduate Student Scholarship.

**Competing interests:** The authors have declared that no competing interests exist.

Exposure to adverse stimuli, such as early-life stress (ELS) during critical windows of developmental plasticity, may produce maladaptive changes that can lead to sensory and motor impairments. For example, maternal stress can accelerate maturation of brain circuits regulating emotion and mental health [9] and speech perception [10]. These functional changes are reflected in ELS accelerating neural maturation of the hippocampus [11, 12] along with premature decline in the expression of cell proliferation and differentiation markers in hippocampus and an earlier switch in the N-methyl-D-aspartate receptor (NMDAR) subunit expression leading to precocious developmental suppression of contextual fear [11]. Similarly, elevated plasma glucocorticoid levels due to ELS can accelerate electrophysiological and morphometric biomarkers of hippocampal aging in rats [12]. As the timing of critical periods differs across brain regions [13], it can be argued that ELS disrupts the orchestrated developmental trajectory of brain regions by accelerating development of specific regions that provide essential functions for immediate survival at the cost of functional disconnect or developmental delay of other regions [14, 15].

The timing of critical periods has been most thoroughly studied in the visual cortex [16, 17]. Although the development of visual circuitry begins prior to the onset of vision [18–20], the developmental trajectory of the visual system is particularly responsive to exogenous stimuli and an advantageous model to study environment-brain interactions. For example, complex sensory-motor stimulation promotes eye-opening, visual acuity, and the decline of white matter-induced long-term potentiation [21]. These effects are accompanied by increased brain-derived neurotrophic factor (BDNF) and glutamate decarboxylase expression [21], thus supporting the critical role for glutamate signaling in modulating the critical period for visual development. By contrast, ELS favors neuronal preference of lower spatial frequency and higher temporal frequency and delays the development of ocular dominance columns [15]. Although the physiological, molecular, and behavioural milestones of visual system development have been well-characterized, little is known about how ELS opens and closes critical developmental periods. This knowledge is important to inform potential interventions that can prevent or modify these circuits in adulthood.

The consideration of ELS is particularly important with regards to its lifelong brain and behaviour consequences. Moreover, ELS consequences can be transmitted to generations of offspring and induce similar impacts on remote offspring [22–24]. Thus, the study of ELS and its effects on critical periods brain development will identify developmental milestones and biomarkers that may translate into improved prevention and diagnostics for populations at risk. While most studies use specific types of stressors (e.g., social stressors, restraint, or food deprivation), multidimensional ELS, distinguished by the concomitant occurrence of multiple discrete stressors, may more authentically represent the multifaceted character of stress exposure in nature. Accordingly, multidimensional ELS caused by animal transportation represents an ecologically valid and common stressor in laboratory and livestock animals leading to profound metabolic reprogramming [25], thus potentially impeding reproducibility of preclinical research and wellbeing [26, 27].

Here, we used a multi-omics approach to characterize the behavioural, metabolic, immunohistochemical, and transcriptomic consequences in the visual cortex following multidimensional ELS caused by transportation on postnatal day (P) 12. We show that ELS induces a precocious shift in the critical period for visual cortical plasticity that is associated with characteristic metabolomic and transcriptomic profiles.

## Materials and methods

### Experimental design

One hundred twenty-seven male C57BL/6 mice (*Mus musculus*) were used. Seventy-two of these animals were bred and raised under consistent laboratory conditions at the local

vivarium (Canadian Centre for Behavioural Neuroscience, University of Lethbridge, AB, Canada). While this lineage was designated as "Control" group, their ancestors were obtained from Charles River Laboratories (Charles River Laboratories, Inc., Saint-Constant, QC, Canada) at least five generations ago. Fifty-five animals were shipped directly from Charles River Laboratories and designated as "Stress" group. These pups and their mothers were shipped concurrently from 7:30 AM to 7:30 PM on P12, thus allowing the experiments to assess the impact of multidimensional ELS on both visual developmental milestones and post-critical period function along with other health outcomes. Animals younger than P12 were not subjected to transport stress to minimize the mortality risk due to transportation. The shipment included 3.25-h of ground transportation and a 5-h flight as airfreight.

All pups remained with their mother until weaning at 9.6 grams (at least P21), after which animals were housed in groups of at least two, in accordance with contemporary animal care practices. Mice were housed under a 12:12-h light/dark cycle with light starting at 7:30 AM. The room temperature was maintained at 22˚C. Standard mouse chow food and water were provided *ad libitum*. All procedures were approved by the University of Lethbridge Animal Care Committee in compliance with the guidelines by the Canadian Council on Animal Care (#1506).

In mice, eye-opening under regular developmental conditions occurs from P12-14 and non-stressed animals may be expected to exhibit eye opening during this period, The visual critical period usually occurs from P20-35 [16, 54, 55]. The present experiments focused on these two timepoints as well as late adolescence/early adulthood (P50) to assess both the immediate and long-lasting effects of multidimensional ELS. Control (n = 72) and shipment stress (n = 55) pups were inspected for eye-opening at 3:00 PM from P13 to P15. Visual acuity was assessed in a subset of these animals at 1:00 PM on P20, P28, and P35 using the virtual optomotor system (control, n = 12; stress, n = 17). Similarly, visual development was assessed in a second subset of animals at 1:00 PM on P20 using the visual cliff test (control, n = 9; stress, n = 14). To assess the impact of ELS on brain metabolism, left and right cerebral tissues from these mice were extracted, weighed, and processed for metabolomic profiling by $^1$H NMR spectroscopy in late adolescence/early adulthood (P50). To assess the impact of ELS on global mRNA expression profiles, visual cortices were collected from a third subset of animals at P20, P35, and P50 (n = 4/group) for transcriptomic analyses. Brains were collected from a fourth group of mice at P35 for immunohistochemical analyses to assess abundance of biomarkers indicative of changes in the developmental trajectory of the primary visual cortex (n = 6/group). Procedures were performed by three female experimenters blind to the experimental condition.

## Eye-opening

From P13 to P15, pups (control, n = 72; stress, n = 55) were inspected for eye-opening daily at 3:00 PM. Eye-opening was defined as the initial break in the membrane sealing the lids of both eyes.

## Behavioural testing

**Visual cliff.** The visual cliff apparatus was used to assess depth perception in offspring at 1:00 PM on P20 [28–32]. The apparatus consisted of a clear, open-topped Plexiglass box (41.0 x 41.2 cm square x 30.4 cm high) positioned on the edge of a second box so that half of its base was in contact with a surface (the "shallow region") and half was suspended 30.4 cm above the table (the "deep region"). The border between the shallow and deep regions was referred to as "the cliff". A sheet of checkered fabric was placed between the two boxes and lined the table

surrounding them. Each animal was placed facing the cliff at the centre of the shallow region furthest from the cliff and filmed for 5 min. The video recordings were analyzed for the total time spent in the deep region as a measure of cliff avoidance.

**Virtual optokinetic system.** The virtual optokinetic system (VOS) was used to assess visual acuity at 1:00 PM on P20, P28 and P35 [33]. Briefly, the animal was placed on an elevated platform surrounded by four computer monitors (43.2 x 43.2 cm). A camera recorded the animal from above to track the optokinetic response to a visual stimulus. The visual stimulus consisted of a virtual cylinder covered by a rotating sinusoidal grating of variable phase, contrast and spatial frequency. The position of the animal's head was tracked manually for 5 min, and OptoMotry software (CerebralMechanics Inc., Medicine Hat, AB, Canada) on an Apple computer used the resulting coordinates to adjust the spatial frequency of the grating in response to the animal's point-of-view. Each new trial was accompanied by a reversal in the direction of the grating's rotation. Spatial frequency thresholds were obtained by increasing the spatial frequency of the sinusoidal grating (0.003 cycles/degree) until the animals were no longer able to track the stimulus. The maximum spatial frequency capable of eliciting head tracking indicates an animal's visual acuity.

## Immunohistochemical analyses

**Tissue collection.** Brains were collected at P35 (n = 6/group) for immunohistochemical analysis. Animals were placed under 4% anesthesia immediately prior to receiving an intraperitoneal overdose of sodium pentobarbital (150 mg/kg, Euthansol; CDMV Inc., Saint-Hyacinthe, QC, Canada). Mice were then intracardially perfused with 1M phosphate-buffered saline (PBS) and 4% paraformaldehyde (PFA; Sigma-Aldrich, St. Louis, MO), respectively. Brains were extracted and stored in 4% PFA for 24 hours at 4˚C, after which they were cryoprotected in a 30% sucrose solution at 4˚C (Fisher Scientific, Ottawa, ON, Canada). Brains were sectioned on a freezing microtome (AO Scientific Instruments, Buffalo, NY, USA) at 40 μm.

**Immunofluorescent staining.** Coronal brain sections collected at P35 underwent immunofluorescent staining for perineuronal nets (PNNs). For each brain, a 2:12 series of free-floating coronal sections was washed 6 times in 0.5% PBST for 7 min/wash. Tissue sections were subsequently incubated with Lectin from *Wisteria floribunda* in 1% BSA in 1M PBS for 18 hours at 4˚C (1:100; Sigma-Aldrich, St. Louis, MO, USA). Sections were then washed 6 times in 0.5% PBST for 7 min/wash, incubated with Streptavidin, Alexa Fluor 633 conjugate in 1% BSA in 1.0 M PBS for 2 hours (1:2000; Fisher Scientific, Ottawa, ON), and 1% DAPI was added to the solution 20 min prior to the end of the incubation with secondary antibody (1:1,000). Tissue sections were rinsed 6 times in 1.0 M PBS for 7 min/ rinse and mounted on slides with a fluorescent mounting medium. Specificity of the PNN antibody has been confirmed in mouse tissue [34]. The dilutions were further optimized, and non-specific labelling was controlled for by incubating sections with 1.0 M PBS instead of primary antibody. All negative controls resulted in the absence of immunoreactivity.

**Stereological cell counts.** Cell counting was performed by an experimenter blind to the treatment groups. Images were analyzed for cell number in the primary visual cortex (area V1) using the optical fractionator method [35, 36]. Sections were analyzed using the Stereo Investigator software (MicroBrightfield, Colchester, VT) integrated with a Zeiss Axio Imager M1 microscope (Carl Zeiss AG, Oberkochen, Germany) and a PCO SensiCam (PCO AG, Kelheim, Germany). The boundaries of the primary visual cortex were identified using the Allen Mouse Brain Atlas according to their coordinates (Bregma -3 to -4.5 mm) [37]. After tracing the primary visual cortex using a 2.5x objective, a point grid was overlaid onto each section.

Stereological cell counts were subsequently performed using a 40x objective. The counting variables were as follows: distance between counting frames, 150 x 150 μm; counting frame size, 70 x 70 μm; dissector height, 25 μm; guard zone thickness, 15%. Cells were counted only if they did not intersect forbidden lines.

## Transcriptomic analyses

**Tissue collection.** Mice (n = 4/group) received an intraperitoneal overdose of sodium pentobarbital (150 mg/kg; Euthansol; Merck, QC, Canada). The brains were rapidly removed, and visual cortices from both hemispheres dissected and flash-frozen together (-80˚C) for global mRNA expression profiling. Norgen Biotek's Fatty Tissue RNA Purification Kit (Cat# 36200, Norgen Biotek, Thorold, ON, Canada) was used to extract total RNA.

**mRNA analyses.** mRNA analyses were performed using an Illumina GAIIx genomic analyzer (Illumina 462 Inc., San Diego, CA, USA) with multiplex. Every library was sequenced across 3 separate lanes, with base calling and demultiplexing performed using Illumina CASAVA 1.8.1 under default settings and Mouse—GRCm38 (Ensembl) used as a reference. Sequence and annotation information were downloaded from iGENOME (Illumina). Data from raw counts were uploaded onto R, where initial data exploration and outlier detection were performed using arrayQualityMetrics and DESeq2 bioconductor packages. Raw counts initially underwent normalization and variance stabilization as described in the DESeq2 manual. Hierarchical clustering analysis of transcriptional profiles was performed in an unbiased manner based on the top 100 most variable genes selected from a subset of highly-expressed genes (i.e., higher than the median expression). Clustering analysis was performed using the heatmap.2 function from the gplots package with the default clustering algorithm, with gene expression values displayed as heatmaps. Similarities between samples were also visualized as PCA plots generated using the plotPCA function implemented in DESeq2. Outlier detection and transcriptional profile quality control was performed using the arrayQualityMetrics package.

## Metabolomic analyses

**Sample collection and preparation.** Mice received an intraperitoneal overdose of sodium pentobarbital (150 mg/kg; Euthansol; Merck, QC, Canada). Because previous research suggested hemispheric asymmetries in stressed rodents [38], the left and right cerebral tissues were separately weighed, stored at -80˚C and processed. Left and right visual cortices were pooled for further analysis to satisfy the tissue mass required for NMR-based metabolomics. To isolate water-soluble metabolites for NMR analysis, tissues were subjected to methanol-based protein precipitation and chloroform-based lipid extraction [39]. Specifically, deionized $H_2O$ (4.85 mL/g), 99% methanol (4 mL/g; Sigma-Aldrich, MO, USA), and chloroform (4 mL/g; Sigma-Aldrich, MO, USA) were added to each tissue sample. Samples were homogenized and centrifuged at 1,000 g for 15 min at 4˚C. After centrifugation, 750 μL aliquots of supernatant were transferred to 2.0-mL centrifuge tubes and allowed to evaporate. Subsequently, 600 μL of phosphate buffer was added. A 4:1 ratio of $KH_2-PO_4$:$K_2HPO_4$ buffer was titrated to a pH of 7.4 using 3 M HCl and 0.02% w/v of sodium azide was added. The samples were centrifuged at 12,000 g for 5 min at 4˚C and 550 μL of the supernatant was transferred to a 5-mm NMR tube for NMR analysis.

**NMR data acquisition and processing.** Data were acquired using a 700 MHz Bruker Avance III HD spectrometer (Bruker, Milton, ON, Canada). Spectra were obtained using a Bruker triple resonance TBO-Z probe with the outer coil tuned to the nuclei of $^1H$, $^{31}P$ and $^2H$ and the inner coil tuned to the $^{13}C$ nucleus. The standard Bruker 1-D NOESY gradient water

suppression pulse sequence 'noesygppr1d' was used with a mixing time of 10 ms. Samples were acquired with 128 K data points, a sweep width of 20.5136 ppm, and a recycle delay of 4 s. Total acquisition size was 128 k. The resulting spectra were zero filled to 256 k, line-broadened by 0.3 Hz, Fourier transformed to the frequency domain, phased, and baseline-corrected. Spectral processing used Bruker Topspin software (version 3.2, patch level 6), and spectra were exported to MATLAB (MathWorks, Asheboro, MA, USA) for spectral binning, data normalization, and scaling. Spectra were binned using Dynamic Adaptive Binning [40] the area corresponding to water was removed, and the dataset was normalized using the Constant Sum method [41] and Pareto-scaled. All peaks were referenced to TSP (0.00δ).

**Metabolite identification.** The Chenomx 8.2 NMR Suite (Chenomx Inc., Edmonton, AB, Canada) was used to identify metabolites present in tissue spectra. Metabolite identities were validated using the Human Metabolome Database [42].

## Statistical analyses

Statistical computations pertaining to eye-opening, visual behaviour, and immunohistochemistry were performed using SPSS (version 26.0, SPSS Inc., IL, USA). Mean values ± standard error of the mean (SEM) were calculated for each measure. For eye-opening, the number of eyes open was recorded for each animal (from 0 to 2). Visual cliff and immunohistochemical data were analyzed using independent samples $t$-tests. VOS and eye-opening data were analyzed using two-way mixed ANOVAs, with Group (stress vs. control) as the between-subjects factor and Time (P13, P14, and P15 for eye-opening; P20, P28, and P35 for VOS data) as the within-subjects factor. Greenhouse-Geisser corrected $p$-values and Bonferroni corrections were applied where appropriate. Data pertaining to visual cliff avoidance in stressed animals relative to controls were then related to metabolic outcomes using the Pearson $r$ correlations in MATLAB. A $p$-value of less than 0.05 was chosen as the significance level.

For transcriptomics, raw count data underwent normalization and regularized *log* transformation using statistical routines implemented in the DESeq2 bioconductor package using default settings as described in the user manual [43]. Pairwise comparisons between experimental groups were also performed using DESeq2. mRNAs with false discovery rate adjusted (FDA) $p$-values $<0.1$ were considered differentially expressed. All results are shown as the means ± standard error of the mean (SEM).

For metabolomics, spectral bins were analyzed per group and classified as either significant or non-significant using a decision tree algorithm [44] and a Mann-Whitney U test [45]. A total of 321 spectral bins were included in left cerebrum analyses and 353 bins were included in right cerebrum analyses. All $p$-values were Bonferroni-Holm corrected for multiple comparisons. Variation in spectral data was visualized using Principal Component Analysis (PCA) using Metaboanalyst [46]. MSEA, Pathway Topology Analysis, and Joint Pathway Analysis were performed using Metaboanalyst to identify metabolic and mRNA-based biochemical pathways altered in response to ELS. Pathway analyses identified the most relevant pathways [47] altered both within and across cerebral hemispheres based on the KEGG pathway database (metabolites), the Entrez Gene database (mRNA transcripts), and the Over Representation Analysis selected, using a hypergeometric test [41].

## Results

### ELS accelerates visual system development

A functional test battery was used to determine milestones of visual system development. A significant interaction between Age and Group ($F(1.65, 206.21) = 23.34$, $p<0.001$) and a

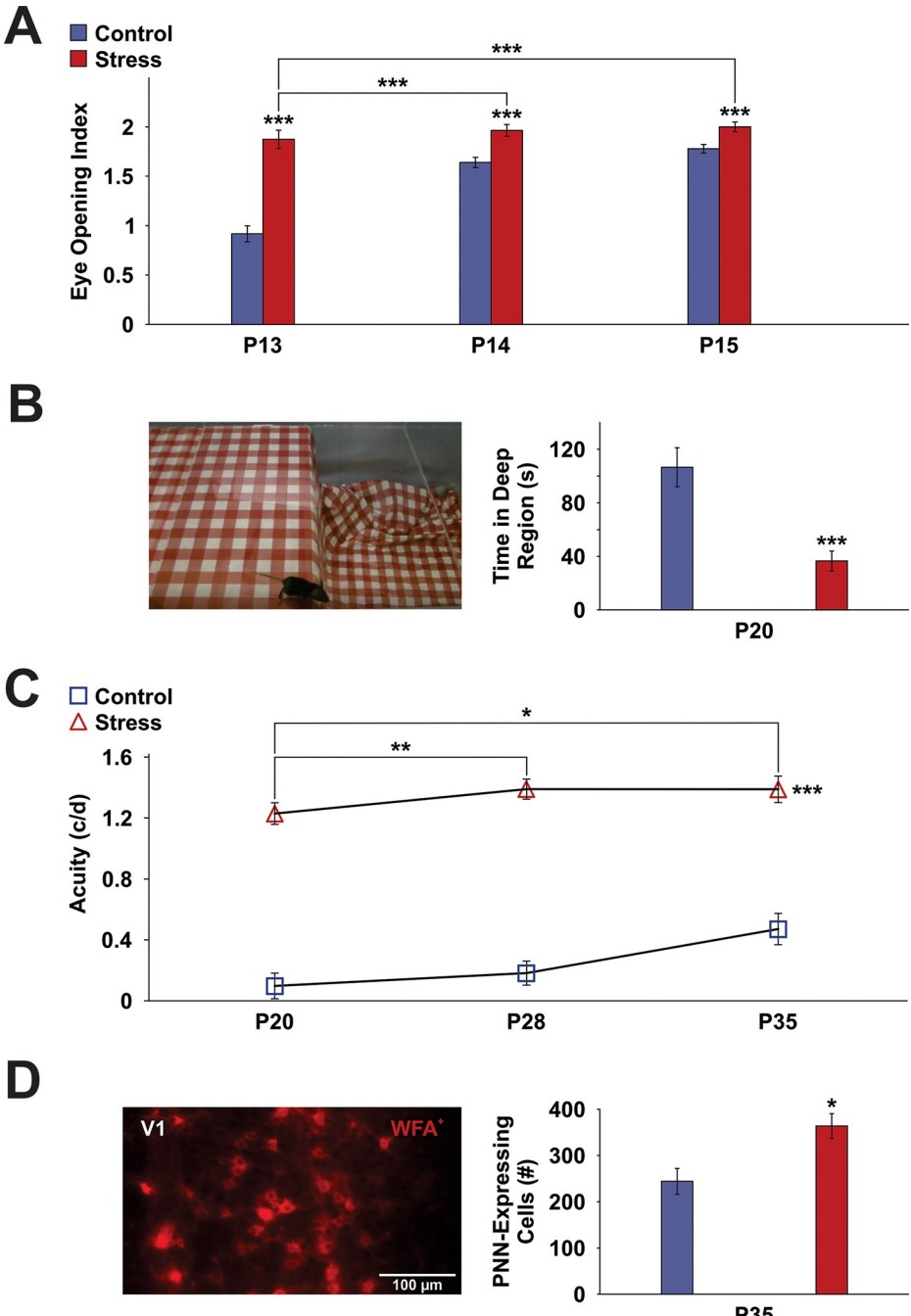

**Fig 1. Multidimensional stress induces precocious development of the visual system.** (**A**) Eye Opening Index throughout early adolescence showing accelerated eye-opening in multidimensionally-stressed animals at P13, P14 and P15. (**B**) Photograph of a mouse in the shallow region of the visual cliff apparatus, observed to measure the precocious development of depth perception. Stressed animals spent significantly less time in the deep region of the apparatus, indicating increased perception of the visual cliff. (**C**) Stressed animals showed improved visual performance relative to non-stressed controls in the visuospatial testing box as measured by the optokinetic reflex. (**D**) Photograph of WFA$^+$ cells, indicating the presence of PNNs, in the primary visual cortex of an adolescent control mouse. Stressed animals showed an increased abundance of PNN-expressing cells at the closure of the critical period for visual system development relative to non-stressed controls. Asterisks indicate significances: $^*p < 0.05$, $^{**}p < 0.01$, $^{***}p<0.001$. Error bars represent ± SEM.

significant main effect of Age for stressed (Fig 1A; $F(1.30, 69.98) = 5.25$, $p < 0.05$) and control ($F(1.67, 118.59) = 45.63$, $p < 0.001$) mice were found during development (P13 to P15), with more eyes open in older mice than in younger ones. Further pairwise comparisons revealed that eye opening significantly increased from P13 to P14 and from P13 to P15 only in non-stressed controls (all $p's < 0.001$). Importantly, eye opening in stress animals exceeded that of controls at P13 ($F(1,125) = 59.77$, $p < 0.001$), P14 ($F(1,125) = 16.75$, $p < 0.001$), and P15 ($F(1,125) = 11.71$, $p < 0.001$). This finding indicates that stress developmentally accelerated eye opening.

In line with accelerated visual system development, stressed animals also showed greater visual cliff avoidance at the start of the critical period for visual cortical plasticity (P20) as well as a significant decrease (M 36.5, SD 27.7) in the time spent in the deep side of the visual cliff relative to controls (M 106.6, SD 43.7) [$t(21) = 4.73$, $p < 0.001$] (Fig 1B).

During development (P20, P28, P35), a significant main effect of Age was found for visual acuity, with older mice displaying higher visual acuity relative to younger mice (Fig 1C; $F(1.49, 40.31) = 8.13$, $p < 0.01$). Further pairwise comparisons revealed that visual acuity significantly increased from P20 to P28 ($p < 0.05$) and from P20 to P35 ($p < 0.01$). Importantly, stress mice overall showed higher visual acuity compared to controls ($F(1, 27) = 149.93$, $p < 0.001$).

## ELS promotes precocious condensation of PNNs

Critical period closure is indicated by the condensation of PNNs. A representative Wisteria floribunda agglutinin (WFA)-stained cross section through the primary visual cortex of a mouse is shown in Fig 1D. Stereological analysis revealed that stress led to an increased abundance of PNN-expressing cells in the primary visual cortex of adolescent mice (Fig 1D). Specifically, the number of PNN-expressing cells in the primary visual cortex was significantly increased in stress mice relative to non-stressed controls [$t(10) = -3.10$, $p < 0.05$]. This observation indicates an early closure of the critical period for visual cortical plasticity in stressed mice.

Transcriptome analysis reflects gene expression changes during visual system development. Global mRNA expression profiling revealed differentially regulated mRNA in response to stress during visual cortex development (see Fig 2 and S2 Table). After adjusting $p$-values using the Benjamini and Hochberg correction [48], global mRNA expression profiles revealed that 13 mRNAs (P20: 3, P35: 7, P50: 3) were differentially expressed in response to stress. These included Mgat5b (upregulated due to stress; FDR $p < 0.05$), Faim2 (upregulated due to stress; FDR $p < 0.01$), Hlf (upregulated due to stress; FDR $p < 0.05$), and Vhl (downregulated due to stress; FDR $p < 0.01$). Other mRNAs that approached significance included Lin7b (FDR $p = 0.099$), Gas6 (FDR $p = 0.099$), and L1cam (FDR $p = 0.063$) (upregulated due to stress) as well as Dusp1 (FDR $p = 0.099$) (downregulated due to stress). Thus, exposure to stress dynamically regulated mRNA expression across development, with unique mRNAs altered in response to stress at P20, P35, and P50.

## ELS perturbs visual cortex metabolism in relation to visual function

To identify down-stream metabolic changes linked to up-stream gene regulation by stress, metabolomic analysis of the left and right primary visual cortices included 321 and 353 spectral bins, respectively. A MW test was applied to identify which features led to differences between the groups, revealing 247 (76.9% of spectral bins) and 305 (86.4% of spectral bins) significantly altered in left and right visual cortex, respectively. In both left and right visual cortices, unsupervised PCA scores plots showed clear group separation, with principal component 1 being equal to 61.9% and 84.0% of the total variance and principal component 2 being equal to

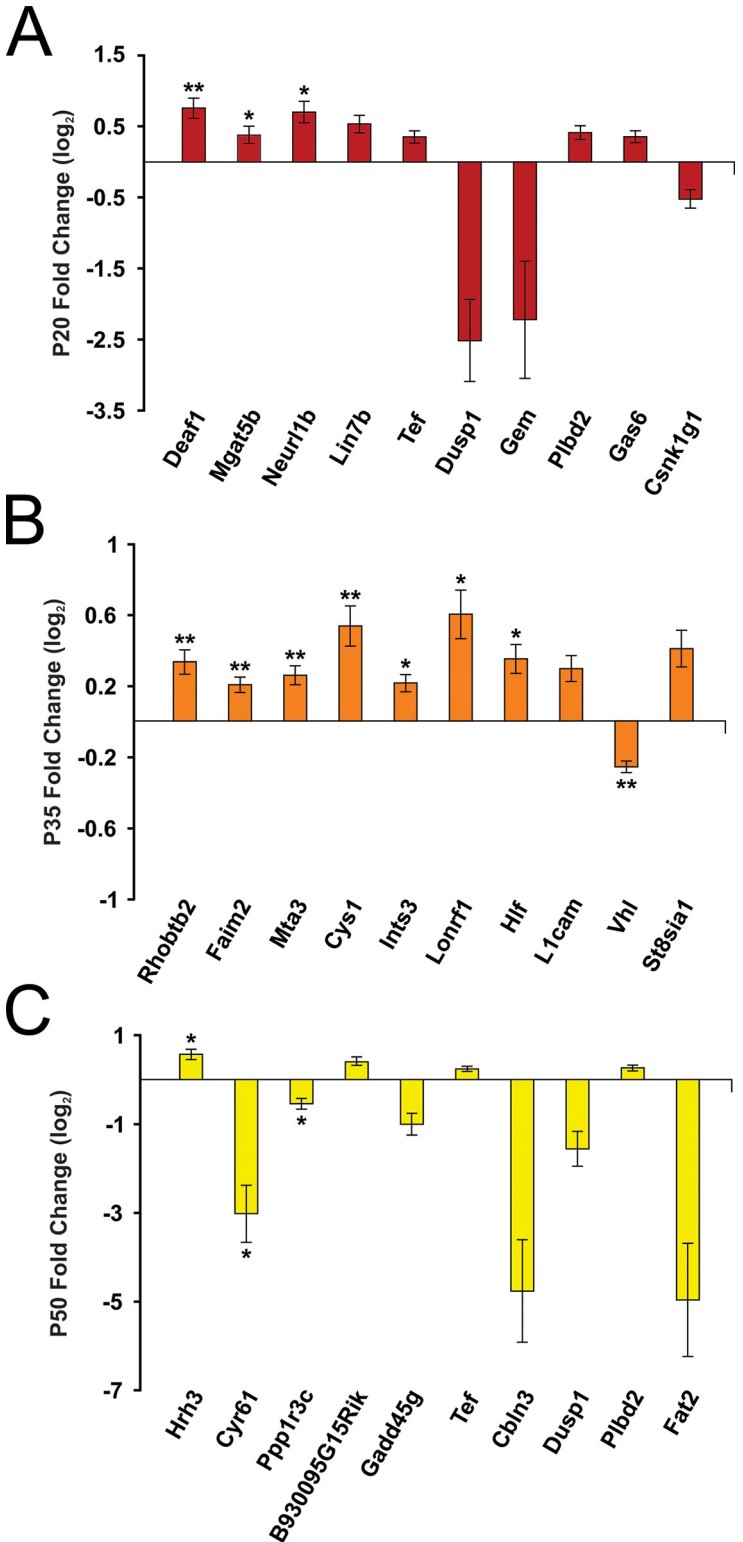

**Fig 2. ELS programs mRNA expression profiles across development.** Fold changes ($\log_2$) of mRNA in P20 (**A**), P35 (**B**), and P50 (**C**) stressed animals in reference to controls. Note that EPS led to up- or down-regulation of unique mRNA transcripts across development. Asterisks denote significances (FDR-adjusted; *$p < 0.05$, **$p < 0.01$). Error bars represent ± SEM.

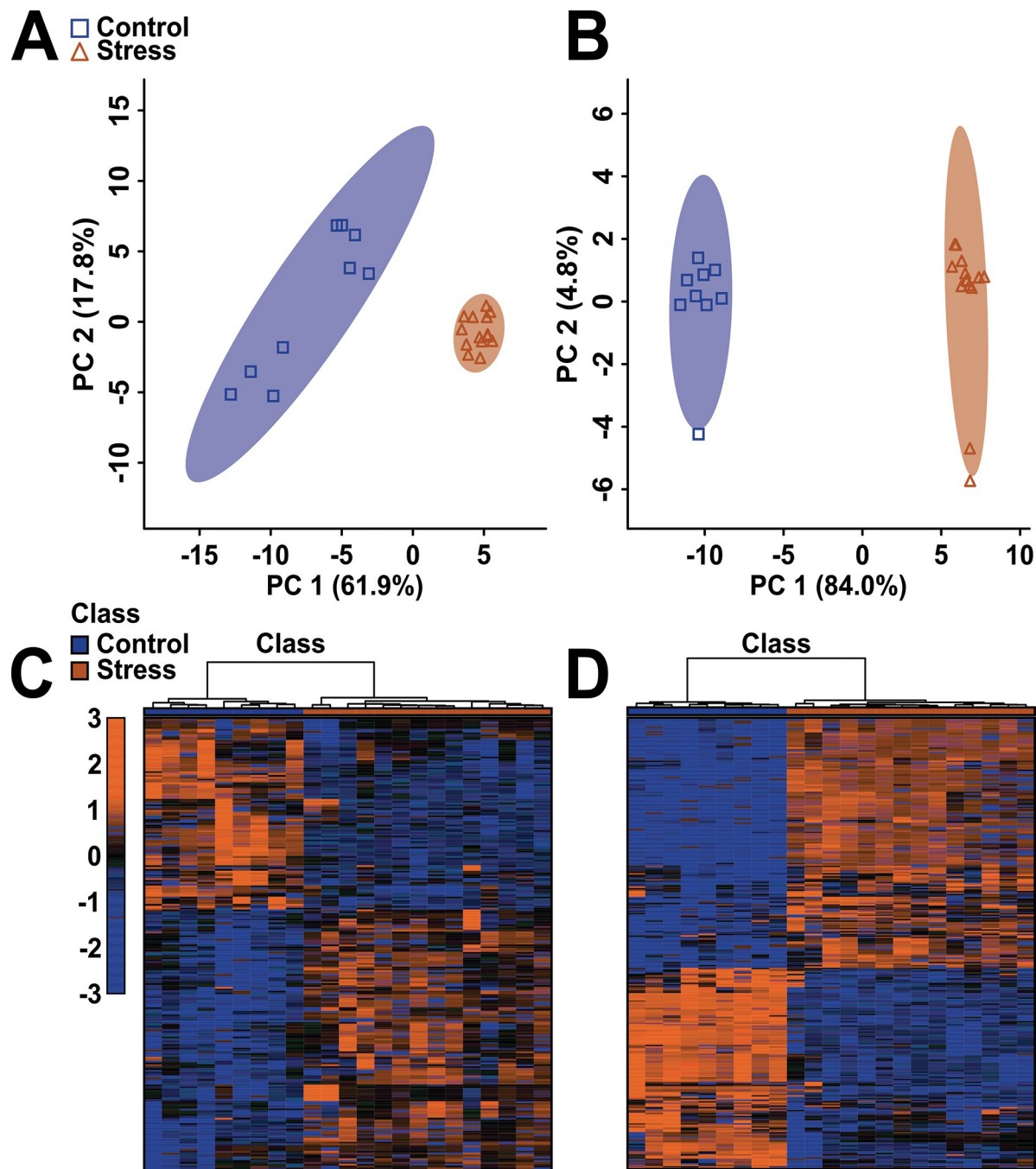

**Fig 3.** PCA scores plots (**A, B**) and heat maps (**C, D**) showing statistically significant separation between adult mice exposed to EPS and controls for both left (**A, C**) and right (**B, D**) cerebra, plotted using a list of metabolites found to be statistically significant by a Mann-Whitney U-test. For the PCA plots (**A, B**), each triangle or square represents one individual under study. The *x*- and *y*-axes show principal components 1 and 2, respectively, with the percentages shown in brackets along each axis indicating the amount of data variance explained by that component. For the heat maps (**C, D**), the *x*- and *y*-axes show the class and individual spectral bins, respectively. These heat maps visually indicate either upregulation or downregulation of the metabolites presented in S1 Table. The dendrogram at the top of each heat map illustrates the results of the unsupervised hierarchical clustering analysis.

17.8% and 4.8% of the total variance, respectively (Fig 3A and 3B). The heat maps presented in Fig 3 [left (Fig 3C) and right (Fig 3D) cortices] illustrate the relative concentrations of metabolites altered as a result of exposure to stress.

**Table 1. Top twenty bins, and their corresponding metabolites, from the left and right cerebrum found to be most significantly altered by stress in a Mann-Whitney U test.** Metabolite regulation is shown as a function of relative concentration in high-EPS animals. Metabolites for which more than one NMR resonance peak was identified as significant are represented as metabolite.1, metabolite.2, . . . metabolite.n. *Indicates metabolites that were significantly altered by stress in both left and right cerebra. The complete list of all significantly altered metabolites can be found in S2 Table.

| Region | Metabolite | NMR Chemical Shift Range of Bin (ppm) | Mann-Whitney U Test | Percent Difference | Regulation by Stress |
|---|---|---|---|---|---|
| | Uracil.1* | 5.253–5.248 | 1.53E-02 | 503.57 | Up |
| | Uracil.2* | 5.258–5.253 | 1.53E-02 | 420.13 | Up |
| | Uracil.3* | 5.414–5.406 | 1.82E-02 | 200.28 | Up |
| | Singlet 3.055 ppm | 3.060–3.049 | 8.25E-05 | −72.52 | Down |
| | 2-Aminoadipate.1* | 2.243–2.234 | 8.25E-05 | 68.96 | Up |
| | Agmatine.1*, Phenylalanine.1*, Taurine.1* | 3.268–3.249 | 8.25E-05 | 62.38 | Up |
| | Adenosine.1* | 8.268–8.259 | 8.25E-05 | −55.10 | Down |
| | Uracil.4* | 5.820–5.814 | 8.25E-05 | 54.35 | Up |
| | Adenosine.2* | 6.119–6.108 | 8.25E-05 | −52.74 | Down |
| Left Cerebrum | Uracil.5* | 5.809–5.803 | 8.25E-05 | 51.82 | Up |
| | Adenosine.3* | 6.108–6.097 | 1.07E-04 | −51.29 | Down |
| | Adenosine.4* | 8.378–8.354 | 8.25E-05 | −48.99 | Down |
| | Nicotinurate.1* | 8.561–8.547 | 8.25E-05 | 44.55 | Up |
| | Histidine.1* | 7.166–7.155 | 8.25E-05 | 44.49 | Up |
| | Glutamate.1* | 2.358–2.348 | 8.25E-05 | −44.31 | Down |
| | Glutamate.2* | 2.348–2.336 | 8.25E-05 | −43.76 | Down |
| | Adenosine.5* | 4.457–4.449 | 8.25E-05 | −43.51 | Down |
| | Formate* | 8.475–8.453 | 8.25E-05 | −43.31 | Down |
| | Choline.1* | 3.492–3.480 | 1.38E-04 | 42.05 | Up |
| | Ethanolamine.1*, Homoserine.1*, Uridine.1* | 3.818–3.811 | 8.25E-05 | 41.81 | Up |
| | Inosine.1 | 6.119–6.108 | 8.25E-05 | −108.20 | Down |
| | Inosine.2 | 6.108–6.097 | 8.25E-05 | −106.69 | Down |
| | Adenosine.1*, Inosine.3 | 8.378–8.354 | 8.25E-05 | −102.02 | Down |
| | Nicotinate.1 | 8.268–8.259 | 8.25E-05 | −98.94 | Down |
| | Methylmalonate.1 | 1.194–1.185 | 8.25E-05 | −98.38 | Down |
| | Uracil.1* | 5.820–5.814 | 8.25E-05 | 91.81 | Up |
| | Uracil.2* | 5.809–5.803 | 8.25E-05 | 90.99 | Up |
| | Inosine.4 | 6.097–6.088 | 8.25E-05 | −87.98 | Down |
| | Phenylalanine.1* | 7.440–7.429 | 8.25E-05 | 86.76 | Up |
| | Leucine.1* | 0.980–0.948 | 8.25E-05 | 86.52 | Up |
| Right Cerebrum | Phenylalanine.2* | 7.352–7.337 | 8.25E-05 | 85.08 | Up |
| | Phenylalanine.3* | 7.429–7.418 | 8.25E-05 | 82.90 | Up |
| | 3-Phenylpropionate.1*, Phenylalanine.4* | 7.337–7.324 | 8.25E-05 | 82.12 | Up |
| | Tryptophan.1 | 7.166–7.155 | 8.25E-05 | 80.51 | Up |
| | N-Acetylaspartate.1* | 7.992–7.946 | 8.25E-05 | −79.16 | Down |
| | Guanosine.1 | 4.457–4.449 | 8.25E-05 | −78.45 | Down |
| | Tryptophan.2 | 7.210–7.199 | 8.25E-05 | 78.38 | Up |
| | Adenosine.2* | 6.088–6.074 | 1.07E-04 | −76.58 | Down |
| | Homocysteine.1 | 1.056–1.046 | 8.25E-05 | 76.53 | Up |
| | Valine.1* | 1.046–1.035 | 8.25E-05 | 76.26 | Up |

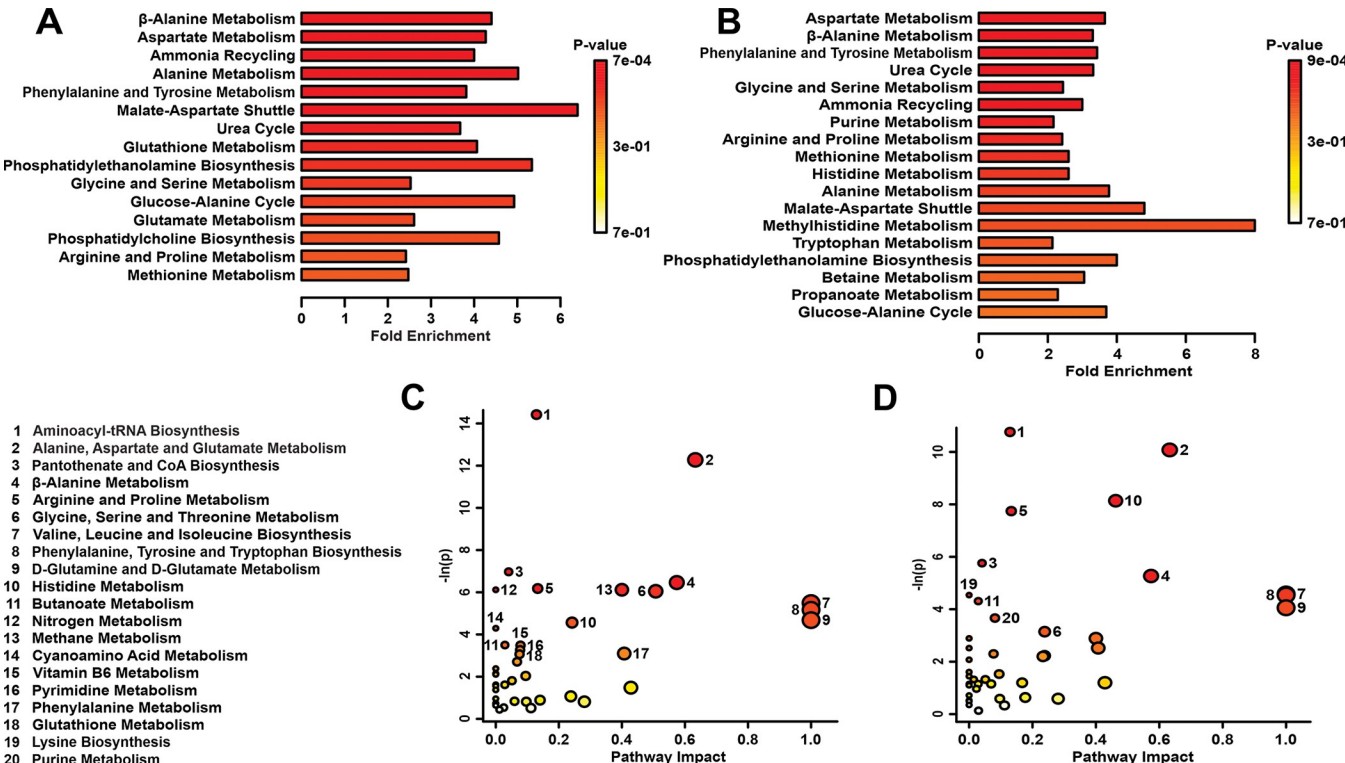

**Fig 4.** (**A, B**) MSEA plot in adult mice exposed to EPS. (**C, D**) Metabolomic Pathway topology analysis showing all matched pathways according to *p*-values from pathway enrichment analysis and pathway impact values in left (A, C) and right (B, D) cerebra of adult animals. A higher value on the *y*-axis indicates a lower *p*-value. The *x*-axis gives the Pathway Impact. Only metabolic pathways with $p < 0.05$ are labeled. This figure was created using the lists of metabolites identified as significant in a Mann-Whitney U test.

Table 1 summarizes the percent differences of stress-induced changes in cortical metabolites for the top twenty bins that found to be significantly altered based on a Mann-Whitney U test (Table 1). The complete list of significantly altered bins and their corresponding metabolites for both hemispheres is provided in S2 Table. The left visual cortex revealed changes in uracil (503.57% difference), 2-aminoadipate (68.96), and adenosine (–55.10). Here, stress up-regulated 27/48 (56.3%) significantly altered metabolites relative to controls, with the remaining metabolites being down-regulated (S2 Table). The right visual cortex revealed changes in inosine (–108.20% difference), nicotinate (–98.94), and methylmalonate (–98.38). Here, stress up-regulated 32/63 (50.8%) significantly altered metabolites relative to controls, with the remainder being down-regulated (S2 Table). Overall, stress up-regulated 24 metabolites and down-regulated 19 metabolites in both left and right hemispheres (S2 Table).

Metabolite set enrichment analysis involved a genome-wide network model of mouse metabolism to identify metabolite sets altered by stress (Fig 4A and 4B). In left cerebra, stress most significantly affected β-alanine metabolism ($p<0.001$), aspartate metabolism ($p<0.001$), and ammonia recycling ($p<0.01$) (Fig 4A). In right cerebra, stress most significantly altered aspartate metabolism ($p<0.001$), β-alanine metabolism ($p<0.01$), and phenylalanine and tyrosine metabolism ($p<0.01$) (Fig 4B). Additionally, energy metabolism was altered as indicated by pathway topology analysis. In the left cerebra, pathways of aminoacyl-tRNA biosynthesis ($p<0.001$), alanine, aspartate, and glutamate metabolism ($p<0.001$), valine, leucine, and isoleucine biosynthesis ($p<0.01$), phenylalanine, tyrosine, and tryptophan biosynthesis ($p<0.01$), and D-glutamine and D-glutamate metabolism ($p<0.01$) were significantly altered (Fig 4C). In the

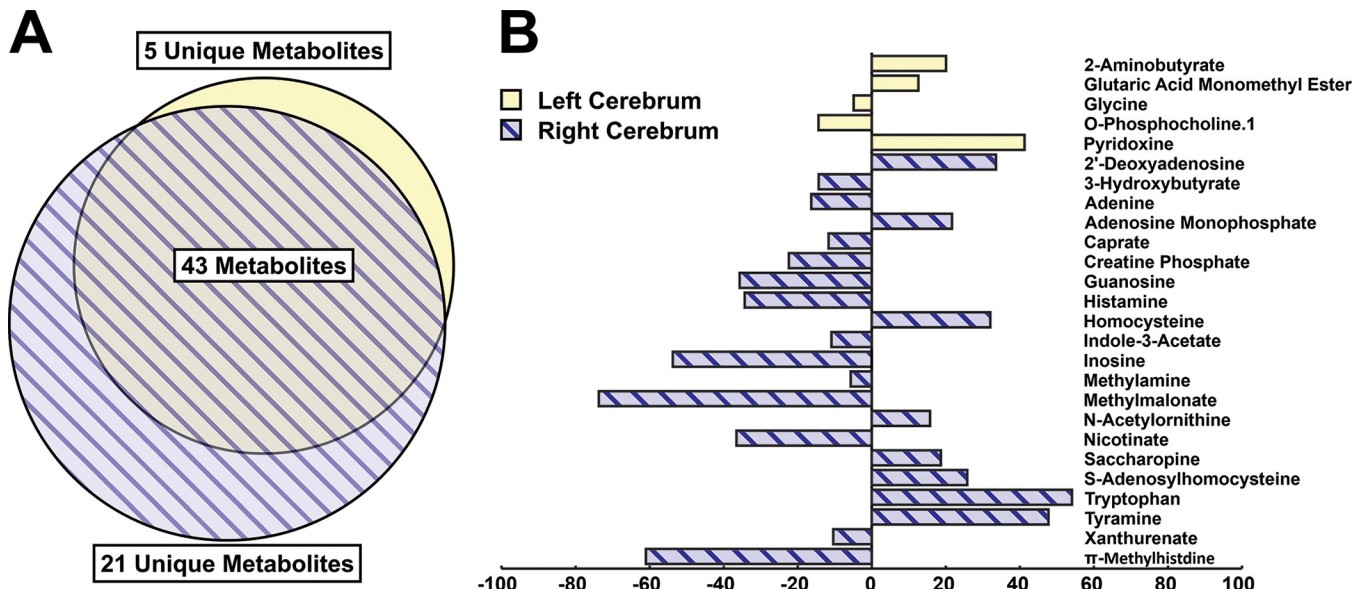

**Fig 5. Multidimensional EPS induces brain lateralization in metabolic profiles.** Venn diagram profiling metabolites changed either uniquely or across both left and right cerebral hemispheres in stressed animals relative to controls (**A**). Percent differences of individual metabolites significantly altered in response to stress ($p < 0.05$; **B**).

right cerebra, significant changes were found in aminoacyl-tRNA biosynthesis ($p<0.001$), alanine, aspartate, and glutamate metabolism ($p<0.001$), histidine metabolism ($p<0.001$), and valine, leucine, and isoleucine biosynthesis ($p<0.01$) (Fig 4D). Cerebral metabolic profiles were altered in a lateralized manner, with 5 (7.2%) & 21 (30.4%) unique metabolites significantly altered in left and right cerebra, respectively (Fig 5A and S3 Table). The left hemispheres revealed unique changes in pyridoxine (41.27% difference), 2-aminobutyrate (28.28), and o-phosphocholine (–20.24), while top metabolites altered in right cerebra included methylmalonate (–98.38% difference), π -methylhistidine (–60.98), and inosine (–108.20) (Fig 5B). Metabolomic pathway analysis performed using metabolites unique to either left or right hemispheres revealed vitamin B6 metabolism ($p<0.05$) significantly altered in left cerebra, while purine ($p<0.01$) and histidine ($p<0.05$) metabolism were significantly altered in right cerebra.

Joint Pathway Analysis integrating differentially-expressed metabolites and mRNA transcripts revealed stress-induced changes across both hemispheres in ABC transporters ($p<0.001$), protein digestion and absorption ($p<0.001$), and aminoacyl-tRNA biosynthesis ($p<0.001$; S4 Table). Processes uniquely altered in the left hemisphere included glutathione metabolism ($p<0.01$) and thiamine metabolism ($p<0.05$), while joint pathways enriched in the right hemisphere included the cAMP signalling pathway ($p<0.01$), tyrosine metabolism ($p<0.05$), and tryptophan metabolism ($p<0.05$) (S4 Table).

Correlational analysis was used to identify the relationship of metabolites to functional visual system development. Pearson $r$ correlations revealed that 172 (53.6% of all spectral bins) and 279 (79.0% of all spectral bins) of altered metabolites were significantly correlated with visual cliff avoidance (S5 Table). Notably, the time spent in the deep side of the visual cliff was negatively related to the relative concentrations of aspartate, glutamate, and tyrosine ($r = –$ 0.78, $p<0.001$) and positively related to glutamate ($r = 0.75$, $p<0. 001$) and tyrosine ($r = 0.73$, $p<0.001$) in left cerebra (Fig 6A–6C and S5 Table). Similarly, the time spent in the deep side of the visual cliff was negatively related to β-alanine ($r = –0.76$, $p<0.001$) and positively related to

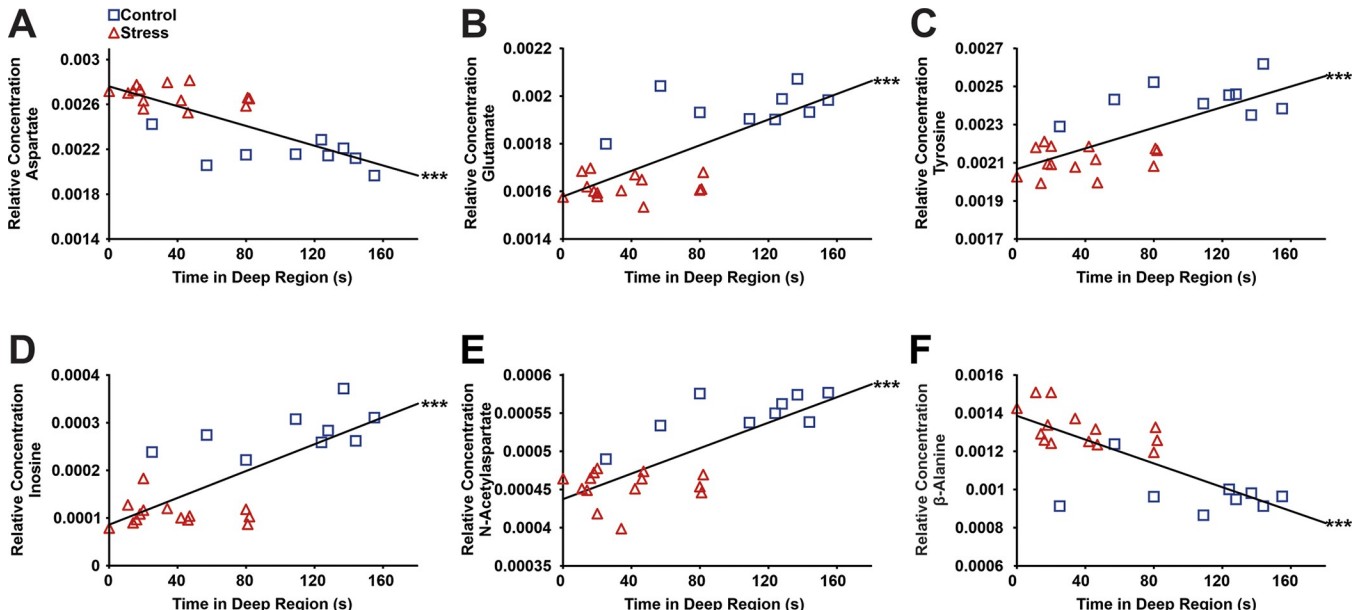

**Fig 6.** Pearson correlations to assess the relationship between precocious development of depth perception (i.e., time spent in the deep region of the visual cliff apparatus) and the relative concentrations of (**A**) aspartate, (**B**) glutamate, (**C**) tyrosine, (**D**) inosine, (**E**) N-acetylaspartate, and (**F**) β-alanine in either left (**A, B, C**) or right (**D, E, F**) cerebra. There were negative correlations between the time spent in the deep region and aspartate ($r = -0.78$, $p = 0.000012$) and β-alanine ($r = -0.76$, $p = 0.000027$), indicating that precocious visual development was linked to higher aspartate and β-alanine concentrations. There were positive correlations between the time spent in the deep region and glutamate ($r = 0.75$, $p = 0.000043$), tyrosine ($r = 0.73$, $p = 0.000077$), inosine ($r = 0.75$, $p = 0.000044$), and N-acetylaspartate ($r = 0.77$, $p = 0.000020$), indicating that precocious visual development was linked to lower glutamate, tyrosine, inosine, and N-acetylaspartate concentrations.

inosine ($r = 0.75$, $p < 0.001$) and N-acetylaspartate ($r = 0.77$, $p < 0.001$) in right cerebra (Fig 6D–6F and S5 Table).

## Discussion

The perinatal environment plays a critical role in shaping brain development and lifetime sensory and behavioural processing [49, 50]. Here, we show that ELS induced by transportation of mice during a vulnerable developmental period, accelerates the trajectory of cerebral development based on (i) the occurrence of precocious eye development and visual function; (ii) earlier cellular condensation of PNNs as indicators of premature critical period closure; and (iii) altered gene expression and hemispheric metabolic asymmetries in primary visual cortex linked to an aberrant developmental phenotype and topographical desynchronization of brain development. Major metabolites regulated by ELS belong to metabolic pathways involved in aminoacyl-tRNA biosynthesis, suggesting a causal pathway how early adverse experiences may promote precocious brain development and functional maturation.

Accelerated visual system maturation and precocious development of depth perception in ELS mice was indicated by their early eye opening, preference for the "shallow side" of the visual cliff, and superior visual acuity throughout the critical period for visual cortical plasticity. While future studies may benefit from measuring eye opening at the supplier's facility in tandem with other treatment groups, these observations provide support for the postulation that ELS causes aberrant developmental trajectories that lead to behavioural impairments in later life [51–53]. The larger scale and permanence of anatomical changes distinguish developmental critical periods from plasticity in adulthood [5], emphasizing the potential impact of ELS on developmental programming of sensory function in later life.

In the present study, precocious visual behaviours throughout the critical period for visual system development in stressed animals were accompanied by increased abundance of PNN-expressing cells in the primary visual cortex at P35, the age of visual critical period closure in the mouse [54, 55]. At the closure of a critical period, parvalbumin-positive cells and their proximal neurites are enclosed by chondroitin sulfate proteoglycan (CSPG)-containing PNNs. Along with myelin factors, these CSPGs bind to the Nogo receptor [56] which then acts along with immune cells to inhibit neurite growth and plasticity [57, 58]. Thus, PNNs can act as a structural molecular "brake" to critical period plasticity, and the appearance of PNNs indicates a closure of the critical period for visual system development [17, 59]. Since WFA fluorescence staining signals rapidly increase in visual cortical neurons from P10-28 as a hallmark of visual system development [60], we reasoned that a significant increase in PNN-positive cells during this period would indicate precocious development. Altogether, these behavioural and immunohistochemical findings indicate that ELS causes an abnormal, accelerated cellular developmental trajectory of the visual cortex along with precocious visual behaviours at the functional level. Notably, CSPGs themselves are a diverse class of proteins including aggrecan, neurocan, and phosphacan, among others. Perfusion-fixed hippocampal and retrosplenial cortical tissues have shown incongruous aggrecan/WFA reactivity [61]. Thus, future work investigating PNN expression patterns as a neurodevelopmental biomarker would benefit from additional intensity-based methods, timepoints, and immunohistochemical approaches, such as colloidal iron binding, to ameliorate the effects of potential methodological confounds and capture the full range of PNN diversity. Additionally, co-staining PNN-positive neurons with cytoplasmic markers may better depict PNN localization. The present data suggest that ELS reprioritizes developmental trajectories of specific brain regions, thus potentially resulting in a topographical desynchronization of brain development. While this may reduce inter-regional connectivity, the disruption of parvalbumin-positive interneurons and their associated PNNs may also contribute to circuit instability [62]. These events have been implicated in schizophrenia, autism, and other psychiatric disorders [62–64]. Accordingly, early life stressors such as parental separation [65] and social isolation [66, 67] were shown to lead to parvalbumin abnormalities in hippocampus and prefrontal cortex, and along with increased PNNs these changes can exacerbate anxiety and hyperactivity [68–70]. The limbic system in particular is sensitive to adverse early-life experiences, with both rats [71] and mice [11] showing an accelerated transition to enduring, mature fear memories coinciding with the premature appearance of PNNs in the amygdala [59]. Moreover, exposure to ELS may induce a shift in critical periods and derail developmental trajectories, raising the risk of behavioural disorders and psychiatric complications in later life [72]. Metabolomics provides a robust tool for risk prediction of these disorders as it directly reflects cellular physiology induced by ELS-induced upstream epigenetic programming [73, 74]. Metabolomic pathways altered by ELS generate a unifying pathogenetic framework that provide insights into risk factors and potential therapeutic targets [75].

Behavioural observations on the visual cliff revealed that ELS resulted in the precocious development of depth perception, as reflected by an increased preference for the shallow region. Precocious visual behaviours seen in stressed animals correlated with the relative concentrations of aspartate, glutamate, and tyrosine in left cerebral tissues as well as inosine, N-acetylaspartate, and β-alanine in right cerebral tissues. In both left and right cerebral hemispheres this correlation was strongest with aspartate, with upregulated relative concentrations of cerebral aspartate in postnatally stressed animals. Aspartate has been shown to modulate N-methyl-D-aspartate receptor (NMDAR)-dependent signaling [76]–specifically, aspartate acts as an endogenous NMDAR agonist by triggering currents through its interaction with each of the two NR2A-D receptor subunits [77]. Interestingly, increased aspartate levels in the brain have been shown to rescue hippocampal age-related deterioration of synaptic plasticity in

mice [77], suggesting a global cerebral desynchronization of synaptic plasticity neuromodulators, with some significantly increased and others significantly decreased in response to stress.

Several metabolites that underwent a significant change in percent difference are involved in mediating plasticity at both the cellular and behavioural levels. Abnormal levels of metabolites involved in gating synaptic plasticity, such as adenosine and inosine, may disrupt the sequential timing of developmental critical periods [78], thus contributing to a topographical desynchronization of brain development and subsequent risk of stress-induced adverse mental health outcomes. Adenosine, which was downregulated in left and right adult cerebral tissues, is known to play an important role in the regulation of neuronal excitability as well as low-frequency synaptic transmission by activating membrane receptors [79–81], while inosine, which has been shown to induce the expression neuronal growth associated proteins and axon extension in models of focal brain injury [82–85], was uniquely downregulated in the right cerebral hemisphere. Moreover, inosine modulates the release of glutamate and serotonin via adenosine receptors and has previously been identified as a potential diagnostic biomarker for depression [86]. Thus, in addition to shifting the critical period for visual system plasticity forward, a stress-induced lateralized imbalance of the inosine-adenosine equilibrium may both attenuate adult neuroplasticity and result in an increased predisposition to affective disorders.

MSEA was used to identify patterns of metabolite concentration changes in a biologically meaningful framework [87]. The most significant pathway altered in left cerebral tissues was β-alanine metabolism, with 7 metabolite hits, and this pathway was also the second most altered in the right cerebellum. β-Alanine, a structural hybrid between α- and γ-amino acid neurotransmitters [88], was upregulated in response to stress. Previously, β-alanine was found to be elevated during the early and late phase of memory retrieval in the Morris water maze, suggesting that this metabolite may play a role in learning and memory [89].

Both MSEA and pathway topology analysis indicated that glutamate metabolism was altered in both hemispheres of shipped animals. Glutamate, an endogenous metabotropic glutamate receptor (mGluR) agonist [90], was significantly downregulated in the cerebra of adult animals exposed to postnatal stress. In the primary visual cortex, cAMP levels are increased by stimulation of mGluRs, which correlate strongly with visual plasticity. Because glutamate is an mGluR agonist, it is integral in modulating the developmental critical period for the visual system [16, 90, 91]. Additionally, Joint Pathway Analysis revealed the cAMP signaling pathway was significantly enriched in right cerebra only, suggesting this signaling cascade may modulate the relationship between glutamate-mediated developmental plasticity and its upstream epigenetic correlates in a lateralized manner.

Multiple metabolites altered in response to stress are involved in energy metabolism. Aberrant energy metabolism contributes substantially to the association between disrupted PV-positive circuits throughout development and later adverse health outcomes. Indeed, a characteristic feature of fast-spiking PV-positive cells is their high metabolic demand, which results in an abundance of reactive oxygen species [92]. The malate-aspartate shuttle, a key mechanism for the transfer of reducing agents (NADH) from the cytosol into the mitochondria for oxidative phosphorylation [93–95], was significantly altered in response to ELS, suggesting the presence of abnormal stress-related mitochondrial energy metabolism. Glutamate is a critical component of this shuttle, and the depletion of cerebral glutamate pools by ammonia may be responsible for malate-aspartate shuttle disruption in the brain [93]. This concept gained further support when the addition of glutamate and ammonia to primary cultures of neurons and astrocytes was found to normalize the malate-aspartate shuttle [96]. Thus, abnormalities in ammonia-mediated cerebral glutamate levels represent an important type of dysregulation in energy metabolism and provide key insights to the mechanisms linking abnormal glutamatergic signaling to stress-associated adverse health outcomes [97].

Consistent with MSEA results and metabolite percent differences, pathway topology analyses for both left and right cerebral tissues revealed changes in D-glutamine and D-glutamate metabolism. Additionally, a significantly altered pathway with one of the highest impact values was valine, leucine, and isoleucine degradation. In agreement with previous studies on the metabolomics of ELS [41], this draws attention to a possible metabolic dysregulation of BCAAs in stressed animals. Abnormal BCAA metabolism has been implicated in the pathogenesis of several chronic conditions including insulin resistance, diabetes, and obesity [98], and may play a mechanistic role in the association between ELS and an increased incidence of chronic disease risk in later life.

Stress also altered the aminoacyl-tRNA biosynthesis pathway, which is central for biosynthesis and kinetics of mRNA translation [99, 100]. Normally, mischarged tRNAs are cleaved by aminoacyl-tRNA synthetases via the actions of a domain distinct from the aminoacylation domain [101, 102]. Dysregulation of aminoacyl-tRNA synthetases results in increased mischarged tRNAs as well as an intracellular accumulation of misfolded proteins in neurons [101, 102]. Interestingly, transcriptomic analyses also revealed a significant downregulation of Vhl, functioning to recruit protein targets for ubiquitination and subsequent proteasomal degradation via the von Hippel-Lindau ubiquitination complex, in late adolescence [103]. Thus, impaired protein degradation mediated by epigenetic programming and altered downstream metabolomic pathways may together compromise the fidelity of cerebral protein translation as well as structural and functional neuronal differentiation [102, 104–106]. Altogether, the broad effects of stress on the metabolomic landscape likely reflect the complex processes spurring the ELS-induced neurodevelopmental shift in the critical period for visual cortical plasticity.

The down-stream metabolic responses reflect up-stream gene expression regulation. Transcriptomic changes in visual cortex revealed dynamic responses to ELS in a focused cohort of neurodevelopmentally-relevant genes, consistent with literature identifying similarly focused ELS-induced changes in the prefrontal cortex [107]. The gene *Mgat5b* encodes a beta-1,6-N-acetylglucosaminyltransferase, which functions in the synthesis of complex cell surface N-glycans [108] that are central to intercellular signaling. In addition, the gene for the Fas apoptotic inhibitory molecule *Faim2* is critical to regulation of apoptosis and neuroprotection [109], thus potentially influencing visual function by regulating neuronal survival and synaptic pruning during brain development. Moreover, the gene for hypoxia-inducible factor *Hlf* is a transcriptional regulator for genes during brain development and involved in biological homeostasis [110]. Finally, the *Vhl* gene encodes the VCB-CUL2 complex that regulates protein degradation and assumes a central role in visual symptoms of the Von Hippel-Lindau disease, a cancer syndrome characterized by vascular tumors affecting the retina, brain and other organs. Mutation in the VHL gene can disrupt cellular processes involving transcriptional regulation, extracellular matrix formation, and apoptosis [111, 112].

The mechanisms underlying the association between early adverse experiences and lifelong disease risk involve complex gene-environment interactions [113]. Accordingly, epigenetic and transcriptomic regulations play a key role in the lifelong effects of early experience-dependent brain plasticity. Indeed, several mRNAs that were differentially expressed in stressed visual cortices are recognized mediators of brain development and neurodevelopmental disorders, including Deaf1 in memory deficits and anxiety-like behaviours [114], Faim2 in obsessive compulsive disorder [115–117] and Hrh3 in schizophrenia [118–120]. Altogether, these results suggest multidimensional ELS may program changes in mRNA expression that predispose an organism to adverse mental health outcomes later in life.

Cumulative multigenerational stress has previously been found to cause a dominance shift toward the right hemisphere, thus provoking a heritable pattern of lateralized behaviours [38].

In other studies, hemispheric specialization has also been associated with motor preferences, language processing, and affective response [121]. To elucidate the potential downstream mechanisms underpinning stress-induced brain lateralization, we examined the hemisphere-specific effects of ELS on brain metabolism. The present findings corroborate our previous work on hemispheric differences linked to ancestral stress, demonstrating a stress-induced shift to right hemispheric dominance [38]. The present findings support this observation through profound metabolic changes affecting the right cerebrum compared to the left side. Even though Ambeskovic et al. (2017) considered multigenerational stress, the first generation exposed to prenatal stress showed a slight functional shift towards the right hemisphere [38] which may reflect a widely altered cellular metabolism. Furthermore, ELS may trigger mechanisms that contribute to neuronal injury, such as a decline in neurotrophin expression [122, 123] or de-regulated microtubule-associated protein 2 (MAP-2), thus impairing synaptogenesis and neurite outgrowth [122, 124]. Additionally, ELS may alter testicular androgen secretion that influences brain development [125] and potential functional asymmetries, such as handedness [38]. Nevertheless, in both left and right cerebral tissues, stress most drastically altered pathways involved in aminoacyl-tRNA biosynthesis. Thus, while there was some lateralization in the specific metabolites altered across cerebral hemispheres, aggregate stress-induced metabolic changes led to a partially shared cerebral phenotype. Accordingly, previous work demonstrated abnormal brain development, functional diversification, and impaired aminoacyl-tRNA biosynthesis in response to early adverse experiences [102]. In addition, ELS represents a significant physiological stressor that may directly accelerate brain developmental trajectories [126], retinal pathologies [127], advance age of menarche [128] and other developmental milestones.

The present findings provide novel insights into the mechanisms underlying perinatal stress-induced disease vulnerability by linking adverse early-life experiences to an aberrant developmental trajectory and altered metabolic phenotype. The findings show that transportation of mice during a vulnerable developmental period induces multidimensional ELS leading to profound changes in brain plasticity and behaviour [25]. Thus, shipping animals from a vendor may affect the reproducibility and reliability of preclinical research [26, 27]. The observed transcriptomic and metabolic changes likely reflect up-stream potentially heritable epigenetic modifications that manifest as downstream alterations in the metabolome and behaviour [126, 129]. In contrast to the impaired visual acuity/binocular matching induced by early-life visual deprivation [130], it can be argued that stress-induced precocious sensory development enhances immediate survival in a dangerous environment but may come at the expense of synchronized brain development with potentially maladaptive consequences in later life [131]. Further insights into the mechanisms underlying ELS-induced disease prevalence along with clearly recognizable metabolic biomarkers of psychiatric disorders open new personalized medicine strategies for early detection and prevention of experience-dependent adverse health outcomes.

## Supporting information

**S1 Table. mRNA expression fold change (log$_2$) with raw and adjusted *p*-values.** Included are FDR-adjusted values with $p<0.1$ in order of smallest adjusted *p*-values. ‡ Indicates mRNA transcripts that were differentially expressed in more than one time-point.
(DOCX)

**S2 Table. Left and right cerebrum metabolites found to be significantly altered by stress in a Mann-Whitney U test.** Metabolite regulation is shown as a function of relative concentration in high-EPS individuals. Metabolites for which more than one NMR resonance peak was

identified as significant are represented as metabolite.1, metabolite.2, . . . metabolite.n. * Indicates metabolites that were significantly altered by stress in both left and right cerebra.
(DOCX)

**S3 Table. Cerebrum metabolites found to be significantly altered by stress in a lateralized manner.** Metabolites found to be uniquely altered in either the left or right cerebral hemisphere using a Mann-Whitney U test. Metabolite regulation is shown as a function of relative concentration in high-EPS animals. Where more than one NMR resonance peak was identified as significant, the magnitude of change reported is the average percent difference of all relevant NMR resonance peaks.
(DOCX)

**S4 Table. mRNA-Metabolite Joint Pathway Analysis visualizing significant genes and metabolites enriched in pathways across left and right cerebra of adult animals.** ‡ Indicates pathways altered uniquely in either left or right cerebral hemispheres.
(DOCX)

**S5 Table. Left and right cerebrum metabolites found to be significantly correlated to precocious visual behaviour.** Pearson correlations were used to assess the relationship between behaviours indicative of precocious visual behaviour (i.e., less time spent in the deep region of the visual cliff apparatus) and relative concentrations of metabolites found to be significantly altered by stress in a Mann-Whitney U test. Positive correlations indicate that precocious development of depth perception was linked to lower metabolite concentrations, while negative correlations indicate that precocious development of depth perception was linked to higher metabolite concentrations. Metabolites for which more than one NMR resonance peak was identified are represented as metabolite.1, metabolite.2, . . . metabolite.n. † Indicates metabolites that were significantly correlated to precocious visual behaviour in both left and right cerebra.
(DOCX)

## Acknowledgments

The authors would like to thank Mr. Michael Opyr for assistance with both the data processing scripts in Matlab and the spectrometer, as well as the University of Lethbridge for the use of the Magnetic Resonance Facility.

## Author Contributions

**Conceptualization:** Janet Poplawski, Gerlinde A. S. Metz.

**Data curation:** Janet Poplawski, Gerlinde A. S. Metz.

**Formal analysis:** Janet Poplawski.

**Funding acquisition:** Gerlinde A. S. Metz.

**Investigation:** Janet Poplawski.

**Methodology:** Janet Poplawski, Tony Montina.

**Project administration:** Tony Montina, Gerlinde A. S. Metz.

**Resources:** Tony Montina, Gerlinde A. S. Metz.

**Software:** Tony Montina.

**Supervision:** Tony Montina, Gerlinde A. S. Metz.

**Validation:** Tony Montina.

**Visualization:** Janet Poplawski.

**Writing – original draft:** Janet Poplawski.

**Writing – review & editing:** Janet Poplawski, Tony Montina, Gerlinde A. S. Metz.

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
