## [Decision Letter · Decision Letter 0]

16 Sep 2024

PONE-D-24-30070Early Life Stress Shifts Critical Periods and Causes Precocious Visual Cortex DevelopmentPLOS ONE

Dear Dr. Metz,

Thank you for submitting your manuscript to PLOS ONE. After careful consideration, we feel that it has merit but does not fully meet PLOS ONE’s publication criteria as it currently stands. Therefore, we invite you to submit a revised version of the manuscript that addresses the points raised during the review process.

**ACADEMIC EDITOR: **

After careful consideration by 2 Reviewers and an Academic Editor, all of the critiques of both Reviewers must be addressed in detail in a revision to determine publication status. If you are prepared to undertake the work required, I would be pleased to reconsider my decision, but revision of the original submission without directly addressing the critiques of the Reviewers does not guarantee acceptance for publication in PLOS ONE. If the authors do not feel that the queries can be addressed, please consider submitting to another publication medium. A revised submission will be sent out for re-review. The authors are urged to have the manuscript given a hard copyedit for syntax and grammar.

We look forward to receiving your revised manuscript.

Kind regards,

Stephen D. Ginsberg, Ph.D.

Section Editor

PLOS ONE

 Journal Requirements: When submitting your revision, we need you to address these additional requirements. 1. Please ensure that your manuscript meets PLOS ONE's style requirements, including those for file naming. The PLOS ONE style templates can be found at https://journals.plos.org/plosone/s/file?id=wjVg/PLOSOne_formatting_sample_main_body.pdf and https://journals.plos.org/plosone/s/file?id=ba62/PLOSOne_formatting_sample_title_authors_affiliations.pdf 2. Thank you for stating the following financial disclosure: "This research was funded by the Natural Sciences and Engineering Research Council of Canada (NSERC) Grants #05628 and #00031 (GM). JP was supported by an Alberta Innovates Graduate Student Scholarship." Please state what role the funders took in the study.  If the funders had no role, please state: ""The funders had no role in study design, data collection and analysis, decision to publish, or preparation of the manuscript."" If this statement is not correct you must amend it as needed. Please include this amended Role of Funder statement in your cover letter; we will change the online submission form on your behalf. 3. In the online submission form, you indicated that "The data underlying the results presented in the study are available from Janet Poplawski (janet.poplawski@gmail.com)." All PLOS journals now require all data underlying the findings described in their manuscript to be freely available to other researchers, either 1. In a public repository, 2. Within the manuscript itself, or 3. Uploaded as supplementary information.This policy applies to all data except where public deposition would breach compliance with the protocol approved by your research ethics board. If your data cannot be made publicly available for ethical or legal reasons (e.g., public availability would compromise patient privacy), please explain your reasons on resubmission and your exemption request will be escalated for approval.  4. Please amend either the abstract on the online submission form (via Edit Submission) or the abstract in the manuscript so that they are identical. 5. Your ethics statement should only appear in the Methods section of your manuscript. If your ethics statement is written in any section besides the Methods, please move it to the Methods section and delete it from any other section. Please ensure that your ethics statement is included in your manuscript, as the ethics statement entered into the online submission form will not be published alongside your manuscript. 6. Please include captions for your Supporting Information files at the end of your manuscript, and update any in-text citations to match accordingly. Please see our Supporting Information guidelines for more information: http://journals.plos.org/plosone/s/supporting-information.

Reviewers' comments:

Reviewer's Responses to Questions

**Comments to the Author**

1. Is the manuscript technically sound, and do the data support the conclusions?

Reviewer #1: Yes

Reviewer #2: Partly

2. Has the statistical analysis been performed appropriately and rigorously? 

Reviewer #1: Yes

Reviewer #2: I Don't Know

3. Have the authors made all data underlying the findings in their manuscript fully available?

Reviewer #1: Yes

Reviewer #2: No

4. Is the manuscript presented in an intelligible fashion and written in standard English?

Reviewer #1: Yes

Reviewer #2: Yes

5. Review Comments to the Author

Reviewer #1: Review Metz et al., PlosONE 2024

Here Poplawski et al., examine the lasting behavioral and molecular consequences of early life stress experience with a focus on critical period development of the visual system. The authors use a paradigm of early life stress (ELS) comparing mice raised undisturbed under standard vivarium conditions through critical period development, to mice that undergo shipping from a commercial vendor at age P12. In this paradigm, the authors show that mice experiencing ELS show accelerated maturation of visual function, including accelerated eye opening, and visual acuity. The authors describe excessive deposition of perineuonal nets (PNNs) marked by staining with lectin WFA, a molecular correlate known to mark the end of critical period plasticity. To gain further mechanistic insights into the lasting effects of ELS the authors conduct mRNA sequencing of visual cortex and metabolic profiling using NMR. The authors show ELS experience drives lasting changes in visual cortex mRNA expression and differential levels of various metabolites with further insights into left and right cortex asymmetry. The authors conclude overall that ELS drives an acceleration of visual system development and premature opening and closing of the critical period plasticity.

The results of the main figures are clearly presented. The results section appropriately describes the findings and associated statistical analysis. The discussion section makes a good case to support the main conclusions and provides thoughtful insights from the mRNA sequencing and metabolic profiling. The overall conclusions of the authors are supported by the data presented. The work represents an area of high importance given the clear emerging roles of early life adverse experience in the onset and risk of psychiatric illness. This work will serve to stimulate further advances in this area of high medical importance. The mRNA and metabolomics data may also serve to stimulate further hypothesis on the mechanisms by which ELS drives lasting changes in brain function and behavior.

Of note, it is important that the authors disseminate the findings of this study for the additional reason that the specific ELS paradigm, involving shipping peri-natal mice from a commercial vendor, is of high relevance to any labs using mouse models and where mice are routinely purchased from commercial vendors for experiments. It will be very important for any labs purchasing mice from vendors to know that mice of vulnerable developmental ages may express long lasting phenotypes, introducing critically important confounds into the experimental design. In this regard it would be interesting and important to determine the window(s) of vulnerability from this type of stress experience, how much younger or older can such changes be observed?

The main criticism of this work is that the authors are not able to provide mechanistic links between accelerated visual system development, deposition of PNNs, and altered expression of mRNA or metabolites. In this regard the paper can be described as descriptive. The data provided offers potential insights into underlying molecular mechanisms that could be examined in further detail on subsequent studies.

Main critique.

Mechanistic links between altered mRNA and metabolites with aspects of visual system function (eye opening, acuity) or anatomy (PNNs) are not provided. Speculative mechanisms are explored in the discussion. Language use in discussion is appropriately moderated as speculative. This reviewer considers further elucidation of mechanisms as outside the scope of the current work, but clearly an area for future studies to focus.

Minor concern

1. The authors can do more to make the metabolomics data easier for readers to understand and appreciate. It is recommended that the (extensive) supplementary tables be revised. Some things to consider, the authors should group metabolites in supp table 2 that were found to be altered by “upregulated” and “downregulated”, at present the metabolites are simply listed alphabetically (maybe?) with “up” and “down” interspersed making it difficult for readers to appreciate overall trends. Same for supp fig4 regarding “positive” and negative” correlation, this would be easier for readers if the two directions were separated.

It would also be useful to have a separate table to describe metabolites that were found to be significantly altered in both left and right hemisphere including direction of change. Fig 5A venn diagram depicts overlap of metabolites between left and right, but the identity and direction of changes of these metabolites is difficult/impossible to obtain from supp Table 2. Shared hemispheric changes in metabolites are depicted in Supp Table 3, but this table is not specific to metabolites so something similar to this would be useful for readers. Table 3 could also include direction of change if possible to increase the value of this table.

2. The lack of bregma numbers lowers the accuracy of comparisons between the control and ELS groups in Figure 1. Please include details of the anatomical coordinates/criteria for histology analysis depicted in Fig 1D.

3. It would have been useful to implement other staining techniques, besides WFA, to improve the validity of PNNs cell counting findings. Consider some description of the different PNN components (proteins/glycans). The PNN literature is ever expanding and it is becoming clear that many of the results obtained from WFA staining do not necessarily capture the range of PNN diversity.

Reviewer #2: The manuscript titled “Early life stress shifts critical periods and causes precocious visual cortex development” uses a mulit-omics approach to address the impact of early life stress on shifting visual critical period. The premise is interesting, and looking at holistic changes at the -omics level in a well-characterized paradigm of visual critical period is a plus. However, the research design and the interpretation of the data with the claims in the text are incongruent.

Questions and comments regarding the manuscript are below.

1. Please define “multidimensional ELS”, as compared to other types of ELS that is typically used in the literature? Do you expect the findings from this study to apply to ELS paradigm in general, or more specifically to this research design?

2. When do the animals raised in Charles River Laboratories, without this “stress” transportation open their eyes? This is a critical control, as the “Control” group in the host institution could have diverged a little bit over the ~ five generations. How can the authors rule out that the standard housing conditions at Charles river didn’t contribute to this early opening phenotype?

3. P50 mice are typically considered “adolescent” rather than “adult” mice. Why do the authors categorize them as adults?

4. Stereological cell counts for PNNs: Most literature for PNN counts use intensity-based measurements to count and categorize PNNs. From the description in the Methods, it is unclear if this was also performed here. This is especially important due to the claim of “precocious shift in critical period”, as PNNs mature from “immature clouds” or wisps of staining to mature and stable extracellular matrix structures. Additionally, since the PNN counts were only done in one time point, how can “precocious” development be claimed? Please see other work, including Krishnan K, Wang BS et al, PNAS 2015; and other references therein that describe PNN maturation over the critical period.

5. The representative image of PNN looks cytoplasmic rather than extracellular. Why? There’s no scale given. The image quality is poor as well.

6. The connection between precocious critical period, binocular vision, visual acuity and PNNs have been well-established. Please explain how precocious eye opening and increase in PNN counts would predict impaired visual acuity and binocular matching, rather than the opposite result in this manuscript. How do the authors reconcile these differences? For example, please see work of Wang BS, Sarnaik R, Cang J in Neuron 2010; Durand S … Hensch TK 2012 et al, and associated work.

7. What accounts for the variation in the “control” dataset in Figure 3, while the “stress” group seems to be more consistent?

8. Figure 3C – figure legend is different color from the actual heatmap in red/green. Please do not use red/green color combination as color-blind people have a hard time accessing this data.

9. Figure 4 – heatmap legend shows “yellow” color while the actual data doesn’t have it. Perhaps in the wrong place? Please reconcile.

10. If “multidimensional ELS” is such a strong disruptor of critical period, it is surprising that the transcriptome is only changed with 13 mRNAs. Is this similar or different from other ELS transcriptome studies? None seem to be cited here.

11. What is the rationale for looking at hemisphere-specific analysis in metabolic studies? More discussion on the interpretation of the hemisphere-specific information would be helpful.

12. Reconciling the -omics data with the implied shift in critical period is unclear. What do the authors think the -omics data in aggregate say about the shift?

13. Were PNNs also measured in a hemisphere-specific manner? PNNs have been shown to have hemisphere-specificity in adult primary somatosensory cortex (Lau BYB et al, eNeuro, 2020), and not in the adolescent primary visual cortex (Emery BA, et al, bioRxiv, 2023).

14. Though there are many relevant citations, some of the newer work on PNNs, visual cortex critical period plasticity, ELS should be updated and cited.

6. PLOS authors have the option to publish the peer review history of their article (what does this mean?). If published, this will include your full peer review and any attached files.

Reviewer #1: No

Reviewer #2: No

---

## [Author Response · Author response to Decision Letter 0]

2 Dec 2024

Reviewer 1 

Here Poplawski et al., examine the lasting behavioral and molecular consequences of early life stress experience with a focus on critical period development of the visual system. The authors use a paradigm of early life stress (ELS) comparing mice raised undisturbed under standard vivarium conditions through critical period development, to mice that undergo shipping from a commercial vendor at age P12. In this paradigm, the authors show that mice experiencing ELS show accelerated maturation of visual function, including accelerated eye opening, and visual acuity. The authors describe excessive deposition of perineuronal nets (PNNs) marked by staining with lectin WFA, a molecular correlate known to mark the end of critical period plasticity. To gain further mechanistic insights into the lasting effects of ELS the authors conduct mRNA sequencing of visual cortex and metabolic profiling using NMR. The authors show ELS experience drives lasting changes in visual cortex mRNA expression and differential levels of various metabolites with further insights into left and right cortex asymmetry. The authors conclude overall that ELS drives an acceleration of visual system development and premature opening and closing of the critical period plasticity.

The results of the main figures are clearly presented. The results section appropriately describes the findings and associated statistical analysis. The discussion section makes a good case to support the main conclusions and provides thoughtful insights from the mRNA sequencing and metabolic profiling. The overall conclusions of the authors are supported by the data presented. The work represents an area of high importance given the clear emerging roles of early life adverse experience in the onset and risk of psychiatric illness. This work will serve to stimulate further advances in this area of high medical importance. The mRNA and metabolomics data may also serve to stimulate further hypothesis on the mechanisms by which ELS drives lasting changes in brain function and behavior.

Of note, it is important that the authors disseminate the findings of this study for the additional reason that the specific ELS paradigm, involving shipping peri-natal mice from a commercial vendor, is of high relevance to any labs using mouse models and where mice are routinely purchased from commercial vendors for experiments. It will be very important for any labs purchasing mice from vendors to know that mice of vulnerable developmental ages may express long lasting phenotypes, introducing critically important confounds into the experimental design. In this regard it would be interesting and important to determine the window(s) of vulnerability from this type of stress experience, how much younger or older can such changes be observed?

The main criticism of this work is that the authors are not able to provide mechanistic links between accelerated visual system development, deposition of PNNs, and altered expression of mRNA or metabolites. In this regard the paper can be described as descriptive. The data provided offers potential insights into underlying molecular mechanisms that could be examined in further detail on subsequent studies.

Main Critique

Mechanistic links between altered mRNA and metabolites with aspects of visual system function (eye opening, acuity) or anatomy (PNNs) are not provided. Speculative mechanisms are explored in the discussion. Language use in discussion is appropriately moderated as speculative. This reviewer considers further elucidation of mechanisms as outside the scope of the current work, but clearly an area for future studies to focus.

Author’s Response:

We thank the reviewer for their laudatory comments and for the constructive feedback. We agree that speculative language was used, and believe this was appropriate given our focus on the downstream consequences of early-life stress exposure. Future directions should further explore the mechanistic underpinnings of the pathways identified in this manuscript.

Minor Concern

1. The authors can do more to make the metabolomics data easier for readers to understand and appreciate. It is recommended that the (extensive) supplementary tables be revised. Some things to consider, the authors should group metabolites in supp table 2 that were found to be altered by “upregulated” and “downregulated”, at present the metabolites are simply listed alphabetically (maybe?) with “up” and “down” interspersed making it difficult for readers to appreciate overall trends. Same for supp fig4 regarding “positive” and negative” correlation, this would be easier for readers if the two directions were separated.

It would also be useful to have a separate table to describe metabolites that were found to be significantly altered in both left and right hemisphere including direction of change. Fig 5A venn diagram depicts overlap of metabolites between left and right, but the identity and direction of changes of these metabolites is difficult/impossible to obtain from supp Table 2. Shared hemispheric changes in metabolites are depicted in Supp Table 3, but this table is not specific to metabolites so something similar to this would be useful for readers. Table 3 could also include direction of change if possible to increase the value of this table.

Author Response:

Thank you for this helpful suggestion. Supplemental Tables 1 (mRNA expression fold changes), 2 (cerebrum metabolite percent differences) and 4 were revised, with metabolites grouped by regulation to facilitate readability. Supplemental Table 3 depicts mRNA-metabolite joint pathway analyses as opposed to individual metabolites. For this reason, this table only contains p-values depicting pathways enriched in stressed animals relative to controls. A separate supplemental table (“Supplemental Table 5”) corresponding to Figure 5 (Venn Diagram) was added to further describe significantly altered metabolites unique to each hemisphere, with the magnitude and direction of changes included. We hope that these additional resources sufficiently will address the previous weakness of our data presentation.

2. The lack of bregma numbers lowers the accuracy of comparisons between the control and ELS groups in Figure 1. Please include details of the anatomical coordinates/criteria for histology analysis depicted in Fig 1D.

Author Response:

Thank you for this comment. Details of the criteria for histological analysis depicted in Figure 1D were included in the revised version of the manuscript (under “Stereological Cell Counts” in “Materials and Methods”).

3. It would have been useful to implement other staining techniques, besides WFA, to improve the validity of PNNs cell counting findings. Consider some description of the different PNN components (proteins/glycans). The PNN literature is ever expanding and it is becoming clear that many of the results obtained from WFA staining do not necessarily capture the range of PNN diversity.

Author Response:

This is an excellent point and has been noted in the “Discussion” as a limitation and a consideration in future directions.

Reviewer 2

The manuscript titled “Early life stress shifts critical periods and causes precocious visual cortex development” uses a multi-omics approach to address the impact of early life stress on shifting visual critical period. The premise is interesting, and looking at holistic changes at the -omics level in a well-characterized paradigm of visual critical period is a plus. However, the research design and the interpretation of the data with the claims in the text are incongruent.

Questions and comments regarding the manuscript are below:

1. Please define “multidimensional ELS”, as compared to other types of ELS that is typically used in the literature? Do you expect the findings from this study to apply to ELS paradigm in general, or more specifically to this research design?

Authors Response: Thank you for your helpful suggestions. Further clarification regarding multidimensional ELS, as well as a brief discussion concerning its relevance to other ELS paradigms, is provided in the introduction section (pages 5-6).

2. When do the animals raised in Charles River Laboratories, without this “stress” transportation open their eyes? This is a critical control, as the “Control” group in the host institution could have diverged a little bit over the ~ five generations. How can the authors rule out that the standard housing conditions at Charles river didn’t contribute to this early opening phenotype?

Author Response: We appreciate this important consideration. On average, murine eye-opening takes place from P12-14, and information regarding eye-opening in non-stressed animals has been added in the “Introduction” section. While future studies can further address this question, we wanted to address transportation as a multidimensional stressor that may itself act as a confound with far-reaching consequences in the life sciences, and chose to explore the context in which this usually happens (i.e., shipment of animals from a supplier to a research institution). While standard housing conditions were maintained across both environments to minimize the risk of this potential confound, the use of two environments was acknowledged as a limitation in the “Discussion”, with a suggestion that future studies aim to include eye-opening measurements of animals at the supplier’s facility in tandem with those made on other treatment groups.

3. P50 mice are typically considered “adolescent” rather than “adult” mice. Why do the authors categorize them as adults?

Author Response: Thank you for raising tis valid point and allow us to clarify this information in the reviewed manuscript. Many studies seem to consider P50 as corresponding to the transition from late adolescence to early adulthood. This clarification has now been made in the manuscript (“Materials and Methods”).

4. Stereological cell counts for PNNs: Most literature for PNN counts use intensity-based measurements to count and categorize PNNs. From the description in the Methods, it is unclear if this was also performed here. This is especially important due to the claim of “precocious shift in critical period”, as PNNs mature from “immature clouds” or wisps of staining to mature and stable extracellular matrix structures. Additionally, since the PNN counts were only done in one time point, how can “precocious” development be claimed? Please see other work, including Krishnan K, Wang BS et al, PNAS 2015; and other references therein that describe PNN maturation over the critical period.

Author Response: Here, binary cell-counting methods were used rather than intensity-based measurements, although supplementing these results with this method in a future study would be valuable. Furthermore, as fluorescence staining signals of WFA staining rapidly increase in visual cortical neurons from PD10-28 as a hallmark of visual system development (Ye & Miao, 2013), we reasoned that a significant increase in PNN counts would correlate with precocious development. As multiple timepoints would no doubt strengthen this hypothesis, both of these points (as well as an overview of the logic underlying the latter) have been incorporated into the “Discussion” of our limitations and future directions consideration. It would be of value that future studies will follow up using additional analytical approaches.

5. The representative image of PNN looks cytoplasmic rather than extracellular. Why? There’s no scale given. The image quality is poor as well.

Author Response: Thank you for highlighting this important feature. The WFA staining has been acknowledged to represent a reliable, robust and widely applied marker for PNNs (Souter & Kwok, 2020), and they appear as lattice-like coatings around neurons. While not directly relevant to the present study, this could have been better illustrated by co-staining with another compound with true cytoplasmic localization, and has been incorporated into our “Discussion” of possible future directions. A scale bar has also been added (Figure 1). 

6. The connection between precocious critical period, binocular vision, visual acuity and PNNs have been well-established. Please explain how precocious eye opening and increase in PNN counts would predict impaired visual acuity and binocular matching, rather than the opposite result in this manuscript. How do the authors reconcile these differences? For example, please see work of Wang BS, Sarnaik R, Cang J in Neuron 2010; Durand S … Hensch TK 2012 et al, and associated work.

Authors Response: Thank you for inspiring an interesting consideration. As described by Wang et al., 2010, “the critical period of binocular matching can be delayed by long-term visual deprivation from birth.” We suggest our multidimensional ELS impacts visual development through an “indirect,” stress-related mechanism, rather than directly through visual deprivation. Indeed, we propose that early-life stress may topographically desynchronize brain development, leading to some areas maturing prior to others so as to maximize reproductive fitness in a suboptimal environment (or, at least, survival up to reproductive age). While this study focused on the phenotypic, behavioural, and metabolic consequences of multidimensional ELS, further research should explore its mechanistic underpinnings. This is a very relevant point, and we have incorporated this information into our “Discussion”. 

7. What accounts for the variation in the “control” dataset in Figure 3, while the “stress” group seems to be more consistent?

Authors Response: We appreciate this note. This trend in data variability is actually not unexpected in these types of analyses. The first step involves incorporating all variables using multivariate modelling, after which a subset of variables are modelled according to the initial results (VIAVC). In this second step, the variation between data points decreases as individuals in the test group are changed in a way that skews comparably after exposure to the independent variable. That is, the variation in the stressed class decreases as the metabolomic changes induced by stress tend to alter the metabolomic landscape conducive to decreased within-group variability when compared to control animals.

8. Figure 3C – figure legend is different color from the actual heatmap in red/green. Please do not use red/green color combination as color-blind people have a hard time accessing this data.

Authors Response: The figures have been revised to utilize a different colour combination – specifically blue/orange, as this colour combination tends to be accessible to individuals with colour-blindness.

9. Figure 4 – heatmap legend shows “yellow” color while the actual data doesn’t have it. Perhaps in the wrong place? Please reconcile.

Authors Response: The p-values of the MSEA results were such that only red bars were present in the significantly-enriched metabolite sets.

10. If “multidimensional ELS” is such a strong disruptor of critical period, it is surprising that the transcriptome is only changed with 13 mRNAs. Is this similar or different from other ELS transcriptome studies? None seem to be cited here.

Authors Response: We agree and indeed the findings highlight the importance of unsupervised assessments to appreciate the range of changes (or lack thereof) across different brain areas. The stress-associated transcriptomic landscape may change according to the stressor, the animal’s developmental stage, and the system being examined; however, it is not uncommon to observe changes in a focused cohort of mRNAs as opposed to a broad effect on the transcriptomic landscape. For example, Orso et al. (2024) found, in the prefrontal cortex, that early-life stress exposure differentially regulated 13 genes in females and 4 in males. However, few studies have looked at brain regions other than the prefrontal cortex, and none have examined the transcriptomic effects of ELS on the visual cortex. To further explore these questions, we are currently performing transcriptomic analyses on motor cortices of animals exposed to ELS. These nuances have been incorporated into our “Discussion”, with add

---

## [Decision Letter · Decision Letter 1]

11 Dec 2024

Early Life Stress Shifts Critical Periods and Causes Precocious Visual Cortex Development

PONE-D-24-30070R1

Dear Dr. Metz,

We’re pleased to inform you that your manuscript has been judged scientifically suitable for publication and will be formally accepted for publication once it meets all outstanding technical requirements.

Kind regards,

Stephen D. Ginsberg, Ph.D.

Section Editor

PLOS ONE

**Comments to the Author**

1. If the authors have adequately addressed your comments raised in a previous round of review and you feel that this manuscript is now acceptable for publication, you may indicate that here to bypass the “Comments to the Author” section, enter your conflict of interest statement in the “Confidential to Editor” section, and submit your "Accept" recommendation.

Reviewer #1: All comments have been addressed

2. Is the manuscript technically sound, and do the data support the conclusions?

Reviewer #1: Yes

3. Has the statistical analysis been performed appropriately and rigorously? 

Reviewer #1: Yes

4. Have the authors made all data underlying the findings in their manuscript fully available?

Reviewer #1: Yes

5. Is the manuscript presented in an intelligible fashion and written in standard English?

Reviewer #1: Yes

6. Review Comments to the Author

Reviewer #1: The authors have addressed my concerns. I have no remaining critiques or suggestions to offer. The authors are to be congratulated for their efforts during revision.

7. PLOS authors have the option to publish the peer review history of their article (what does this mean?). If published, this will include your full peer review and any attached files.

Reviewer #1: No

---

## [Editor Report · Acceptance letter]

13 Dec 2024

PONE-D-24-30070R1 

PLOS ONE

Dear Dr. Metz, 

I'm pleased to inform you that your manuscript has been deemed suitable for publication in PLOS ONE. Congratulations! Your manuscript is now being handed over to our production team.

Kind regards, 

on behalf of

Dr. Stephen D. Ginsberg 

Section Editor

PLOS ONE